# Formative peer assessment in higher healthcare education programmes: a scoping review

Marie Stenberg  , Elisabeth Mangrio, Mariette Bengtsson, Elisabeth Carlson 

► Prepublication history and supplemental material for this paper is available online. To view these files, please visit the journal online (http://dx.doi.org/10.1136/bmjopen-2020-045345).

Department of Care Science, Faculty of Health and Society, Malmö University, Malmö, Sweden

**Correspondence to**
Professor Elisabeth Carlson;
elisabeth.carlson@mau.se

## ABSTRACT

**Objectives** Formative peer assessment focuses on learning and development of the student learning process. This implies that students are taking responsibility for assessing the work of their peers by giving and receiving feedback to each other. The aim was to compile research about formative peer assessment presented in higher healthcare education, focusing on the rationale, the interventions, the experiences of students and teachers and the outcomes of formative assessment interventions.

**Design** A scoping review.

**Data sources** Searches were conducted until May 2019 in PubMed, Cumulative Index to Nursing and Allied Health Literature, Education Research Complete and Education Research Centre. Grey literature was searched in Library Search, Google Scholar and Science Direct.

**Eligibility criteria** Studies addressing formative peer assessment in higher education, focusing on medicine, nursing, midwifery, dentistry, physical or occupational therapy and radiology published in peer-reviewed articles or in grey literature.

**Data extractions and synthesis** Out of 1452 studies, 37 met the inclusion criteria and were critically appraised using relevant Critical Appraisal Skills Programme, Joanna Briggs Institute and Mixed Methods Appraisal Tool tools. The pertinent data were analysed using thematic analysis.

**Result** The critical appraisal resulted in 18 included studies with high and moderate quality. The rationale for using formative peer assessment relates to giving and receiving constructive feedback as a means to promote learning. The experience and outcome of formative peer assessment interventions from the perspective of students and teachers are presented within three themes: (1) organisation and structure of the formative peer assessment activities, (2) personal attributes and consequences for oneself and relationships and (3) experience and outcome of feedback and learning.

**Conclusion** Healthcare education must consider preparing and introducing students to collaborative learning, and thus develop well-designed learning activities aligned with the learning outcomes. Since peer collaboration seems to affect students' and teachers' experiences of formative peer assessment, empirical investigations exploring collaboration between students are of utmost importance.

## Strengths and limitations of this study

► The current scoping review is previously presented in a published study protocol.
► Four databases were systematically searched to identify research on formative peer assessment.
► Critical appraisal tools were used to assess the quality of studies with quantitative, qualitative and mixed-methods designs.
► Articles appraised as high or moderate quality were included.
► Since only English studies were included, studies may have been missed that would otherwise have met the inclusion criteria.

and collaboration between peers are key characteristics. In a peer assessment activity, students take responsibility for assessing the work of their peers by giving (and receiving) feedback on a specific subject.[1] It allows students to consider the learning outcomes for peers of similar status and to reflect on their own learning mirrored in a peer.[2] Peer assessment has shown to support students' development of judgement skills, critiquing abilities and self-awareness as well as their understanding of the assessment criteria used in a course.[1] In higher education, peer assessment has been a way to move from an individualistic and teacher-led approach to a more collaborative, student-centred approach to assessment[1] aligned with social constructivism principles.[3] In this social context of interaction and collaboration, students can expand their knowledge, identify their strengths and weaknesses, and develop personal and professional skills[4] by evaluating the professional competence of a peer.[5] Peer assessment can be used in academic and professional settings as a strategy to enhance students' engagement in their own learning.[6–8] The collaborative aspect of peer assessment relates to professional teamwork, as well as to broader goals of lifelong learning. As argued by Boud *et al*,[1] peer assessment addresses course-specific goals not readily developed otherwise. For healthcare professions, it enhances the

## BACKGROUND

Peer assessment is an educational approach where feedback, communication, reflection

ability to work in a team in a supportive and respectful atmosphere,[9] which is highly relevant for patient outcome and the reduction of errors compromising patient safety.[10] However, recent research has shown that peer collaboration is challenging[11] and that healthcare professionals are not prepared to deliver and receive feedback effectively.[12] This emphasises the importance for healthcare educators to support students with activities fostering these competences. Feedback is highly associated with enhancing student learning[13] and modifying learning during the learning process[14] as a means for students to close the gap between their present state of learning and their desired goal(s). Peer feedback can be written or oral and conducted as peer observations in small or large groups.[8] Further, it is driven by set assessment criteria,[1] which can be either summative or formative, formal or informal. Summative assessment evaluates students' success or failure after the learning process,[15] whereas formative assessment aims for improvement during the learning process.[4 16] According to Black and Wiliam,[15] formative peer assessment activities involve feedback to modify the teaching and learning of the students. The intention of feedback is to help students help each other when planning their learning.[4 17] An informal formative peer assessment activity involves a continuous process throughout a course or education, whereas a formal one is designated to a single point in a course momentum. Earlier research on peer assessment in healthcare education has provided an overview of specific areas within the peer assessment process. For example, Speyer *et al* presented psychometric characteristics of peer assessment instruments and questionnaires in medical education,[18] concluding that quite a few instruments exist; however, these intruments mainly focus on professional behaviour and they lack sufficient psychometric data. Tornwall[12] focused on how nursing students were prepared by academics to participate in peer assessment activities and highlighted the importance of creating a supporting learning environment. Lerchenfeldt *et al*[19] concluded that peer assessment supports medical students in developing professional behaviour and that peer feedback is a way to assess professionalism. Khan *et al*[20] reviewed the role of peer assessment in objective structured clinical examinations (OSCE), showing that peer assessment promotes learning but that students need training in how to provide feedback. In short, the existing literature contributes valuable knowledge about formative peer assessment in healthcare education targeting specific areas. However, there seems to be a lack of compiled research considering formative peer assessment in its entirety, including the context, rationale, experience and outcome of the formative peer assessment process. Therefore, this scoping review attempts to present an overview of formative peer assessment in healthcare education rather than specific areas within that process.

## METHOD

This scoping review was conducted using the York methodology by Arksey and O'Malley[21] and the recommendations presented by Levac *et al*.[22] We constructed a scoping protocol, using a Preferred Reporting Items for

**Table 1** The Population Concept and Context mnemonic as recommended by the Joanna Briggs Institute

| Population | Concept | Context |
|---|---|---|
| Students assessing students | Intervention, rationale, outcome, context and students' and teachers' experience of formative peer assessment | Healthcare education programmes in higher education |

Systematic Review and Meta-Analysis Protocols, to present the planned methodology for the scoping review.[23]

### Aim and research questions

We aimed to compile research about formative peer assessment presented in higher healthcare education. The research questions were as follows: What are the rationales for using formative peer assessment in healthcare education? How are formative peer assessment interventions delivered in healthcare education and in what context? What experiences of formative peer assessment do students and teachers in healthcare education have? What are the outcomes of formative peer assessment interventions? We used the 'Population Concept and Context' elements recommended for scoping reviews to establish effective search criteria (table 1).[24]

### Relevant studies identified

The literature search was conducted in the databases PubMed, Cumulative Index to Nursing and Allied Health Literature, Education Research Complete and Education Research Centre. Search tools such as Medical Subject Headings, Headings, Thesaurus and Boolean operators (AND/OR) helped expand and narrow the search. Initially, the search terms were broad (eg, peer assessment or higher education) in order to capture the range of published literature. However, the extensiveness of the material made it necessary to narrow the search terms and organise them in three major blocks. The following inclusion criteria were applied in the search: (1) articles addressing formative peer assessment in higher education; (2) students and teachers in medicine, nursing, midwifery, dentistry, physical or occupational therapy and radiology and (3) peer-reviewed articles, grey literature (books, discussion papers, posters, etc). Studies of summative peer assessment, instrument development and systematic reviews were excluded. We incorporated several similar terms related to peer assessment in the search to ensure that no studies were missed (online supplemental appendix 1). Furthermore, we consulted a well-versed librarian with experience of systematic search[25] to assist us in systematically identifying relevant databases and search terms for each database, control the relevance of the constructed search blocks and manage the data in a reference management system. No limitation was set for year, all studies indexed in the four databases were included until the last search 28 May 2019.

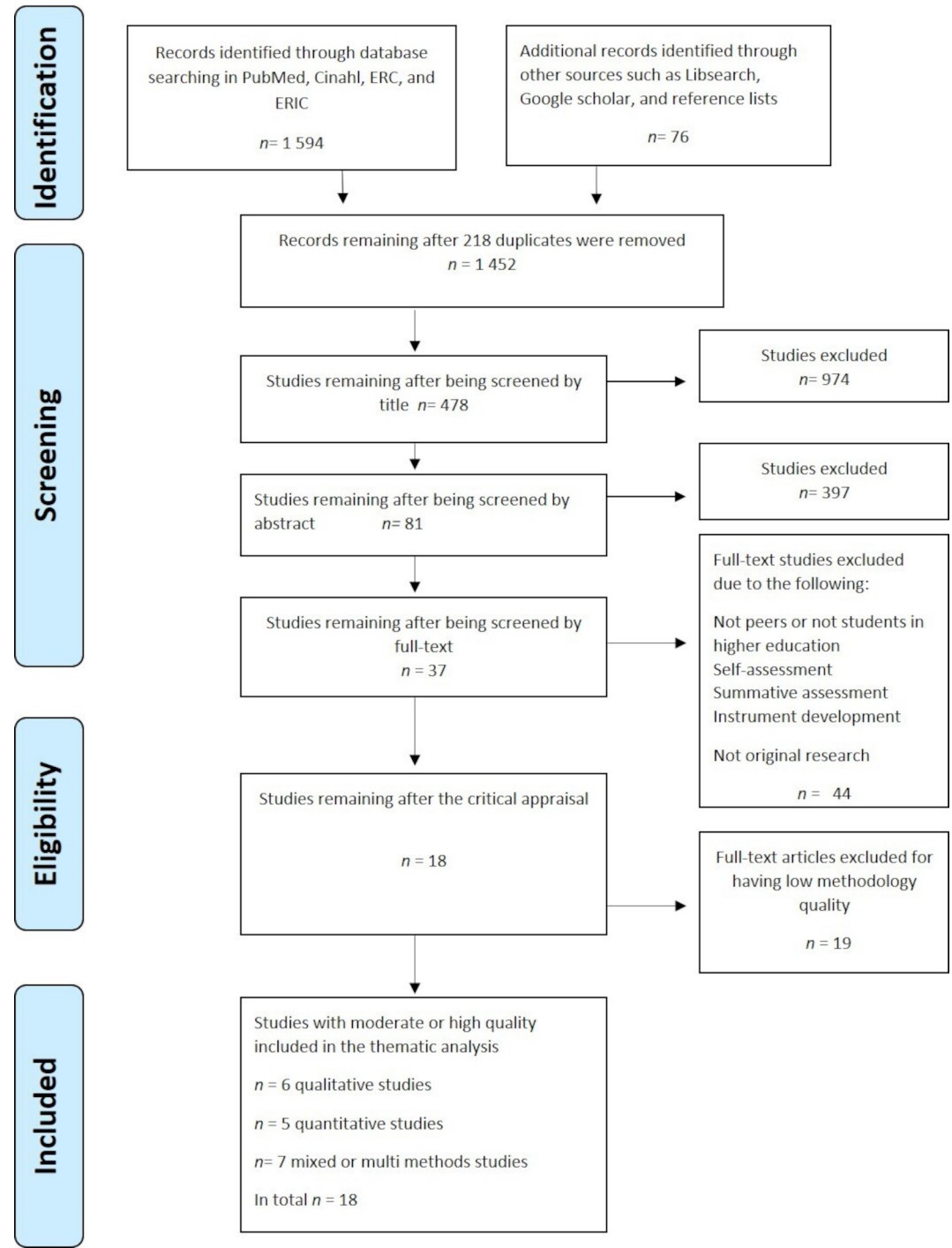

**Figure 1** PRISMA flow chart. ERC, Education Research Centre; ERIC, Education Research Complete; PRISMA, Preferred Reporting Items for Systematic Reviews and Meta-Analyses.

## Study selection

The process of the study selection and the reasons for exclusion are presented in a flow diagram[26] (figure 1). First, the first author (MS) screened all 1452 titles. Second, MS read all the abstracts, gave those responding to the research questions a unique code, and organised them in a reference management system. The reason for inclusion and exclusion at title and abstract level was charted by the first author and critically discussed within the team (MS, EM, MB and EC). An additional

hand search of reference lists was conducted. To cover a subject in full, a scoping review should include search in grey literature.[21 22] Therefore, the grey literature was scoped to find unpublished results by searching Google Scholar, LibSearch and Science Direct. The grey literature mostly contained research posters, conference abstracts, discussion papers and books, but a handsearch revealed original research articles that were added for further screening and appraisal. Finally, the first author (MS) arrived at 81 studies, read them in full-text, and

discussed them with the other three authors (EM, MB and EC).

## Charting the data

We constructed a charting form to facilitate the screening of the full-text studies (online supplemental appendix 2). Out of the 81 studies, 37 met the inclusion criteria and were appraised for quality using Critical Appraisal Skills Programme (CASP).[27] The reason for conducting a crtical appraisal of the studies was to enhance the use of the findings for policy-making and practice in higher healthcare education.[28] To investigate the interpretation of the quality instruments, three members of the research team (MS, EM and EC) conducted an initial test assessment of two randomly selected studies and graded them with high, moderate or low quality. Additional screening tools were used for studies with a mixed methods design[29] and cross-sectional studies[30] not available in CASP. When a discrepancy arose, a fourth researcher (MB) assessed the articles independently without prior knowledge of what the others have concluded. This was followed by a discussion among all four researchers to secure internal agreement on how to further interpret the checklist items and the quality assessments. Consequently, to ensure high quality, the studies had to have a ''yes' answer for a majority of the questions. If 'no' dominated, the study was excluded. Since earlier reports[31] have raised and discussed the importance of ethical issues in systematic reviews, all screening protocols in this review included ethical considerations, as an individual criterion. The first author critically appraised all 37 articles, and 15 articles were divided between the team members (EM, MB and EC) and independently appraised. Nevertheless, during the screening process all 37 articles were critically discussed using the Rayyan system for systematic reviews[32] before final decision for inclusion. By this procedure, all authors agreed on not only which articles to include, but also the reason for exclusion. The critical appraisal resulted in 18 studies with high and moderate quality (table 2).

## Collating, summarising and reporting results

The analysis process followed the five phases of thematic analysis described by Braun and Clarke,[33] with support of a practical guide provided by Maguire and Delahunt.[34] The first phase included familiarising with the data. Therefore, prior to the coding process, we read all the articles to grasp a first impression of the results presented within the included studies.We then conducted a theoretical thematic analysis, meaning that the results were deductively coded,[33] guided by the research questions. We read the results a second time before starting the initial coding. The codes consisted of short descriptions close to the original text. The codes were then combined into themes and subthemes. The themes were identified with a semantic approach, meaning that they were explicit: we did not look for anything beyond what was written.[33] Finally, we constructed a thematic map to present an overview of the results and how the themes related to each other. The results from the studies are presented narratively.

## Consultation

Consultation is an optional stage in scoping reviews.[21] However, since it adds methodological rigour,[22] we presented and discussed the preliminary results and the thematic map with nine academic teachers who are experts within the field of healthcare education and pedagogy. The purpose of the consultation was to enhance the validity of the results of the scoping review and to facilitate appropriate dissemination of outputs.[33] The expert group responded to four questions: Do the themes make sense? Is too much data included in one single theme? Are the themes distinct or do they overlap? Are there themes within themes?[34] The consultation resulted in a revision of a few themes and the way they related to each other.

## Patient and public involvement

No patients or members of the public were involved.

## RESULTS

The 18 included studies were published between 2002 and 2017 in the USA (6), the UK (6), Australia (3), Canada (2) and the United Arab Emirate (1) (table 3). The studies were conducted in medical (12), dental (2), nursing (2), occupational therapy (1) and radiography (1) educations. Six studies were presented in the framework of an existing collaborative educational model.[35–40] Our review revealed that the most frequent setting for formative peer assessment activities is within clinical skilltraining courses,[35 39–47] involving intraprofessional peers. The common rationale for using formative peer assessment is to support students, usually explained by the inherent learning of the feedback process,[35 39 40 43–45 47–51] and to prepare students for professional behaviour and provide them with the skills required in the healthcare professions.[36–38 46–49 52] Table 3 presents the results of the analysis related to the research questions of context, rationale and interventions of formative peer assessment.

The results related to the research questions about the experience of students and teachers and the outcome of formative peer assessment interventions fall within three themes: (1) the organisation and structure of peer assessment activities, (2) personal attributes and consequences for oneself and one's peer relationships and (3) the experience and outcome of feedback and learning.

## The organisation and structure of formative peer assessment activities

In the reviewed studies, students express that the responsibility of faculty is a key component in formative peer assessment, meaning that faculty must clearly state the aim of the peer assessment activity. Students highlight the need to be prepared and trained in how to give and receive constructive feedback.[36 47 50–52] The learning

**Table 2** Overview of included studies

| Author, year, country, and journal | Aim | Design | Participants | Main findings | Quality* |
|---|---|---|---|---|---|
| Arnold, 2005[52]<br><br>USA<br><br>*Journal of General Internal Medicine* | Identify factors that encourage or discourage student participation in peer assessment | Qualitative<br><br>Grounded theory<br><br>Focus groups (16) at two medical schools | n=61, medical students in years 1, 3 and 4 | The characteristics of the peer assessment system and the environment can encourage or discourage participation. Themes: (1) Students' struggle with peer assessment, (2) Characteristics of a peer assessment system, and (3) Environmental factors | Moderate |
| Cho, 2016[41]<br><br>England<br><br>*BMC Medical Education* | Investigate the effect of peer-group size on competency-based skills | Quantitative Cluster Randomised clinical trial | n=115, medical students in year 6 | Smaller groups (4.1) show more active and preferred than large groups (8,1). Group size did not impact scores. | High |
| Chou, 2013[39]<br><br><br><br>USA<br><br><br><br>*Medical Education* | Examine the role of prior peer-learning relationships between students in their delivery and receipt of feedback on clinical communication skills | Mixed method<br><br><br><br>Case–control<br><br><br><br>Descriptive statistics<br><br>Survey, video observations | n=72 medical students in year three with prior peer learning relationships<br><br>n=36 students in control group with no prior peer relationships. | Students with prior peer learning relationships more likely to provide specific corrective feedback than those without prior relationships. No significant difference between groups regarding how feedback was received. | Moderate |
| Cushing, 2011[35]<br><br>UK<br><br><br><br>*Medical Teacher* | Investigate the benefits of formative peer feedback in communication skills and develop a training programme in peer feedback | Mixed method<br><br>Questionnaire (20 items) at two occasions with 6 months in between.<br><br>Focus groups (five medical- and two nurse students) | n=45 medical students in year 1<br><br>n=48 nursing students in year 1 | Students valued the learning opportunity of both being examiner and observer. They preferred more in-depth feedback and feedback from tutors. They expressed anxiety about giving negative feedback to a peer and had mixed views on giving feedback (relaxed or pressured) and its use in clinical placements. | High |
| Elshami, 2017[50]<br><br>United Arab Emirates<br><br>*Radiography* | Assess perception of formative peer assessment | Qualitative<br><br>Action research<br><br>Focus groups (3)<br><br>Content analysis | n=19 (24†) diagnostic radiography students in year 3 | Formative peer assessment gives valuable feedback from same level or more experienced peers. Need for training and detailed rubrics. | Moderate |
| Emke, 2017[38]<br><br>USA<br><br>*Teaching and Learning in Medicine* | Demonstrate that perceptual errors related to professionalism behaviours can be detected early through repeated multisource feedback | Quantitative | n=246 medical students in year 2 | Multiple peer assessments and feedback a tool predictor of unprofessional behaviour. | Moderate |
| Iqbal, 2016[36]<br><br><br><br>Australia<br><br><br><br>*BMC Medical Education* | Explore students' and tutors' perception of key collaborative behaviours that impact collaborative learning and interaction | Qualitative<br><br><br><br>Focus groups (5) with students<br><br>Interviews (8) with teachers<br><br>Thematic analysis | n=22 medical students in year one and two<br><br>n=8 teachers | Being respectful, giving constructive feedback, and being engaged and prepared had positive impact on both learning and group interaction. Passiveness, unreliability, irresponsibility, and condescending attitudes had a negative impact on learning and interaction. Similar results from teachers. | High |

Continued

| Table 2 | Continued | | | | |
|---|---|---|---|---|---|
| Koh, 2010[51]<br><br>UK<br><br>*Nurse Education in Practice* | Explore how academic staff experience, understand and interpret the process of formative assessment and feedback of theoretical assessment | Qualitative<br><br>Phenomenology<br><br>Semi-structured interviews (22)<br><br>Thematic analysis | n=20 academic staff in nurse education | Teachers see themselves as key facilitators and think students prefer teacher feedback. Students are assumed to have the skill to peer assess and give feedback but are unprepared and need support and introduction early in education. Teachers need professional development themselves. | Moderate |
| Mui Lim, 2010[49]<br><br><br><br>Australia<br><br><br>*International Journal of Therapy and Rehabilitation* | Improve students learning through interactive formative assessment and student generated questions | Mixed methods<br><br><br><br>Cohort study<br><br>Evaluation questionnaire | n=115 occupational therapy students in year 1 in 2009 compared with<br><br>n=98 students in 2008 | Significant improvement in exams result from being part of interactive formative assessment, which is beneficial for learning and identifying knowledge gaps. | Moderate |
| Martin, 2014[48]<br><br>Canada<br><br>*Nurse Education Today* | Examine collaborative testing versus traditional test taking with undergraduate nursing students in a nine-station OSCE | Mixed method<br><br><br>Cross-over design<br>Survey<br><br>Focus groups | n=70 nursing students | Significantly higher scores in collaborative testing than in traditional testing.<br><br>Themes: (1) studying more/studying differently, (2)/ cognitive collectivism (3), 'it stuck in my head better' (4), confidence, and (5) practicing how to share knowledge and negotiate. | Moderate |
| Moineau, 2011[42]<br><br>Canada<br><br><br>*Medical Education* | Compare scores and experiences of formative assessment from faculty and senior students during OSCE-examinations | Quantitative<br><br>Cross sectional<br><br><br>Prequestionnaire and postquestionnaire | n=66 medical students in year 2<br><br>n=27 year four student examiners<br><br>n=27 teaching doctors | Students (year 4) assessing students (year 2) with checklists in OSCE-examinations equally assessed compared with faculty members. A positive learning experience expressed from both students and faculty. | Moderate |
| Nofziger, 2010[37]<br><br>USA<br><br><br><br>*Academic Medicine* | Investigate the impact of peer assessment on future professional development and students' experiences | Qualitative<br><br>Questionnaire and narrative comments Frequency count | n=70 medical students in year 2<br><br>n=48 in year 4 | 67% found peer assessment helpful, reassuring, or confirming something they knew; 65% reported important transformations in awareness, attitudes, or behaviours because of peer assessment. Change was more likely when feedback was specific and described an area for improvement. | Moderate |
| Rees, 2002[46]<br><br>UK<br>*Medical Education* | Explore students' perceptions of communication skill assessment | Qualitative<br><br>Focus groups | n=7 medical students in year 1<br>n=7 in year 2<br>n=10 in year 3<br>n=5 in year 4<br>n=3 in year 5 | Year 4 and 5 more positive than younger students. Opportunities to compare communication skills with peers from same level. Learning experience being the assessor. No constructive criticism from peers. Difficult to be objective and to give feedback. | High |
| Satterthwaite, 2008[43]<br><br>UK<br><br>*European Journal of Dental Education* | Investigate if any differences existed between marks given by a peer group and those given by experienced assessors | Quantitative<br><br>Cross sectional | n=65 dental students | No significant difference in grades between experienced examiners and peer group. | Moderate |

Continued

**Table 2** Continued

| Spandorfer, 2014[47] USA *Anatomical Science Education* | Determine whether peer assessment improves students work habits and interpersonal attributes and whether it is accepted by students, focusing on low performing students | Multimethods Paired sample t-test Pearson correlation coefficients Survey-content analysis | n=267 medical students in year 1; follow-up in year 2 | Significant improvement after on-line peer feedback between test 1 and 2. Themes: (1) Initiative, (2) Communication, (3) Respect, (4) Preparation, and (5) Focus. Students prefer anonymous feedback from peers. | Moderate |
|---|---|---|---|---|---|
| Tai, 2016[40] Australia *Advances in Health Science Education* | Investigate students' experience of peer-assisted learning. | Mixed methods Ethnographic Survey, observations, and interviews Thematic analysis | n=10 medical students in year 1 (observed) n=191 students in year 3 (survey) | Observing and giving feedback to peers contributed to learning, but students value feedback from teachers for validation. Students want to preserve social relationships with peers; therefore, feedback is not so constructive. Peers provide a supportive learning environment. | High |
| Tricio, 2016[45] UK *European Journal of Dental Education* | Analyse written feedback provided as a part of a formative and structured peer assessment protocol. | Multimethods Descriptive statistic Thematic analysis | n=40 dental students in year two in pre-clinical skills laboratory n=68 dental students in year 5 in clinic | Year 2 focuses on practical and clinical knowledge; in contrast, year 5 focuses comments on communication, management, and leadership. Year 2 gives more positive comments on peer performance than year 5. | Moderate |
| Vaughn, 2016[44] USA *The American Journal of Surgery* | Evaluate the use, quality, and quantity of peer video feedback and compare peers and faculty feedback. | Quantitative Cross-sectional Paired t-test, Mann-Whitney statistic Survey | n=24 medical students‡ | Significant change in performance across three periods in both groups. Peer feedback group performed better at final assessment than faculty feedback group (not significant). Peers gave higher scores than faculty. No significant differences when using a checklist. | Moderate |

\*High equals majority of items in the critical appraisal tools.
†Twenty-four students included in the intervention, and 19 attended the focus group session.
‡Twelve students received faculty feedback, and 12 students received peer feedback.
OSCE, objective structured clinical examination.

activities need to be well designed and supported by guidelines on how to use them.[35 36 50 52] Otherwise, it could discourage students from participating in the peer activities.[52] Novice students find it difficult to be objective and to offer constructive criticism in a group.[36 46] This emphasises the importance of responsibility from faculty, especially when students are to give feedback on professional behaviour.[52] Some students prefer direct communication with peers when feedback is negative, whereas others think it is the responsibility of faculty.[52] There is some ambiguity regarding whether feedback should be given anonymously or not,[47 52] whether it should bear consequences from faculty or not,[52] whether it should be informal or formal, and whether the peer should be at the same academic level or at a more experienced higher-level.[50 52] Moreover, some students express how they favour small groups[41 49]; as students in small groups are more active than those in large groups.[41] Students and teachers agree that peer assessment should be strictly formative rather than summative.[42 46 52] Teachers see themselves as key facilitators and express that students value feedback from teachers rather than from peers (in terms of credibility).[51] Students express similar sentiments even if they appreciate the peer feedback.[40 42 44 46] However, teachers confirm the need for training and preparing students early in the education, as well as the need for their own professional development to guide students effectively.[51]

### Personal attributes and the impact and consequences for oneself and one's peer relationships

Students generally focus on how peer assessment activities may affect their personal relationships in a negative

**Table 3** Overview and summery of the context, rationale and interventions of formative peer assessment presented in the included studies.

| Contexts | Rationales | Interventions |
|---|---|---|
| Intraprofessional students (17)* | Giving and receiving feeback supports student learning: | Introduction (in workshops): |
| Combination of medical and nursing students (1) | Promotes learning (8) | Preparations in giving and receiving feedback (3) |
| Conducted in the following: | Enhances critical thinking (1) | Introduction of guidelines or checklists to guide the peer assessor (3) |
| Clinical skill labs (11) | Promotes understanding of the assessment process (1) | Introduction of the learning activity (2) |
| Theoretical courses (7) | Develops critical and interpersonal skills (1) | Preparation in communication (1) |
| Combination of theoretical and clinical placement course (1) | Helps identify knowledge gaps (1) | Learning activities focusing feedback on professionalism: |
| Within an educational model as problem-based learning, peer learning or peer assisted learning (7) | Supports low-performance students (1) | Clinical skills (3) |
| | It prepares students for knowledge-related professionalism in the healthcare profession by helping them identify the following: | Collaborative behaviour (2) |
| | Professional and unprofessional behaviour (6) | Clinical reasoning (2) |
| | Clinical competence (2) | Theoretical knowledge (2) |
| | Technical skills (2) | Communication skills (2) |
| | Communication skills (2) | Management skills (1) |
| | Collaborative behaviour (2) | Feedback types: |
| | Evaluative judgement (1) | Face-to-face (7) |
| | It enhances teachers' teaching (1) | Anonymous (5) |
| | It provides cost benefits: | Written (3) or through observations (3) |
| | Students as assessors instead of teachers (2) | Interactive on-line assessment (3) |
| | Students as creators of the learning activities instead of teachers (1) | Grading of the given feedback (1) |
| | | Random peers (8) |
| | | Ability to choose peer (1) |
| | | In small groups <6 (6) |
| | | In large groups >6 (3) |

*Appears in how many of the included 18 studies.

way.[35 37 42 50 52] They express worry over consequences for themselves and their social relationships[37 40 52] as well as feeling anxious that negative feedback given to a peer may affect the grading from faculty.[52] Moreover, students emphasise the importance of enthusiasm and engagement in listening to peers' opinions during their collaboration.[36 47] They mention positive personal attributes and behaviours such as being organised, polite and helpful as supportive for peer collaboration.[36 47] Further, they mention the importance of both a positive and close relationship between students and faculty[52] and a positive culture in the learning environment.[40] While students highlight the impact on and consequences for personal relationships, teachers speak of the importance of respect in formative peer assessment,[36] including respect for each other, the learning activity, and the collaboration and interaction.[36] Further, teachers emphasise the importance of students being self-aware, being well prepared and taking own responsibility for the peer assessment activity.[36]

### The experience and outcome of feedback and learning

According to the students in the reviewed studies, formative peer assessment contributes to developing

the skills needed in practice and in their future profession.[35 36 40 41 48 52] They appreciate the opportunity to give and receive feedback from a peer,[35 36 40 42 47 48 50] and they agree that the feedback they received made them change how they worked[42 48] or how they taught their peers.[47 48] They consider activities such as observation of others' performance as beneficial for learning because they make them reflect on their own performance[35 36 40 41 46 49 50] and help them identify knowledge gaps.[35 40 49] Students with prior experience of peer learning are more likely to provide specific guiding feedback than those without such experiences.[39] Moreover, two studies showed significantly improved test results for students who took part in a peer feedback activity compared with those who did not.[43 49] Further, students thought they could be honest in their feedback and would learn better if the feedback was more in-depth.[35 46] Students at entry level tend to give more positive feedback than senior students; they also focus on practical and clinical knowledge, whereas more senior students focus on communication, management and leadership in their feedback comments.[45] A study exploring what students remember of received feedback points to memories of positive growth, negative self-image and negative attitudes towards classmates. Received feedback sometimes confirmed personal traits the students already knew about.[37] In addition, negative feedback was more likely to result in a change in their work habits and interpersonal attributes.[37] Students expressed some anxiety regarding the usefulness of feedback from low-performing students[40 50] and non-motivated students, which contributes to ineffective interaction and learning.[36 47] Low performing students show lack of initiative, preparation and respect but also improvement in their grades after the peer assessment experience.[47] Furthermore, feedback from peers can be a predictor of a student's unprofessional behaviour; hence, it could be used as a tool for early remediation.[38] In an evaluation of faculty examiners' experience of students' feedback, the faculty express how they consider student feedback to be given in a professional and appropriate way and faculty examiners would have given similar feedback.[42] In an OSCE-examination where a checklist was used, the results showed statistical significance in assessment between faculty examiners and student examiners.[42]

## DISCUSSION

We found that formative peer assessment is a process with two consecutive phases. The first phase concerns the understanding of the rationale and fundament of the peer assessment process for students and faculty members. The results indicate that the rationale is to support student learning and prepare them for healthcare professions. The formative peer assessment activities support students' reflection on their own knowledge and development when mirrored in a peer by alternating the roles of observer and observed.[53 54] It further contributes to skills as communication, transfer of understandable knowledge and collaboration, all

significant core competences when caring for patients and their relatives.[54] For faculty, organising formative peer assessment, can be cost beneficial. This was recently emphasised in high volume classes expressing the reduction of costs with students giving feedback to a peer instead of teachers.[55] Nevertheless, students express the importance of clarifying the aim of the peer assessment activity and the responsibility of the faculty. We recommend faculty to clearly define the activity and explain how it supports student learning and professionalism, especially when students are to provide feedback to each other on sensitive matters, such as unprofessional behaviour. A collaborative activity between students requires trust, and the real intention must be made transparent.[4 56–58] Moreover, to enable student development in line with the learning outcomes, the learning activity needs to be well designed and understood by students.[59–61] However, Casey et al[62] recommended further investigations of how to prepare students for the peer assessment activities.

The second phase concerns the organisation and structure of the formative peer assessment activity, for example, how to give and receive feedback and the complexity of peer collaboration as it affects students' emotions concerning both themselves and their relationship with their peers. This coincides with earlier research emphasising the social factors of peer assessment and the importance for teachers to consider them.[4] Nevertheless, surprisingly, few studies highlight the collaborative part of peer assessment.[4 11] One reason might be that formative peer assessment is often presented as a 'stand alone' activity and not involved in a collaborative learning environment.[8 63] We agree with earlier research[64 65] arguing that peer assessment needs to be affiliated with practices of collaborative learning. Similar implications are presented by Tornwall,[12] who concluded the importance of integrating peer collaboration as a natural approach throughout education to support student development.

## LIMITATIONS

Previous methodological concerns and discussions have been related to the systematic approach of handling grey literature.[66 67] We argue that the grey literature may contribute to a wider understanding of the research area. Nevertheless, when we conducted a critical appraisal of the included studies, the grey literature was excluded due to lack of methodological rigour. Therefore, we recommend considering this time-consuming phase of the methodology in scoping reviews. We further acknowledge that the last search was conducted in May 2019, studies may have been included if an additional search had been provided after this date and in other databases than the ones presented. Further, the current scoping review has not fully elucidated the perspective of teachers and faculty. Few of the included studies highlighted the teachers' perspective why further research is required.

## CONCLUSIONS AND IMPLICATIONS FOR FURTHER RESEARCH

Some have argued that research on peer assessment is deficient in referring to exactly what peer assessment aims to achieve.[68] We conclude that within healthcare education the aim of formative peer assessment is to prepare students for the collaborative aspects crucial within the healthcare professions. However, healthcare education must consider preparing and introducing students to collaborative learning; therefore, well-designed learning activities aligned with the learning outcomes need to be developed. Based on this scoping review, formative peer assessment needs to be implemented in a collaborative learning environment throughout the education to be effective. However, since peer collaboration seems to affect students' and teachers' experience of formative peer assessment, empirical investigations exploring the collaboration between students are of utmost importance.

**Acknowledgements** Special thanks to the members in the expert group for their valuable contribution in the consultation.

**Contributors** MS led the design, search strategy, and conceptualisation of this work and drafted the manuscript. EM, MB and EC were involved in the conceptualisation of the review design, inclusion and exclusion criteria, and critical appraisal and provided feedback on the methodology and the manuscript. All authors give their approval to the publishing of this scoping review manuscript.

**Funding** The authors have not declared a specific grant for this research from any funding agency in the public, commercial or not-for-profit sectors.

**Competing interests** None declared.

**Patient consent for publication** Not required.

**Provenance and peer review** Not commissioned; externally peer reviewed.

**Data availability statement** All data relevant to the study are included in the article or uploaded as online supplemental information. No additional data available.

**ORCID iDs**
Marie Stenberg http://orcid.org/0000-0002-0749-5718
Elisabeth Carlson http://orcid.org/0000-0003-0077-9061

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
