## [Reviewer comments · BMJ Open]

ARTICLE DETAILS

TITLE (PROVISIONAL)	Formative Peer Assessment in Higher Healthcare Education Programs – a Scoping Review
AUTHORS	Stenberg, Marie; Mangrio, Elisabeth; Bengtsson, Mariette; Carlson, Elisabeth

VERSION 1 – REVIEW

REVIEWER	Marit Hegg Reime Western Norway University of Applied Science, Norway.
REVIEW RETURNED	19-Oct-2020

GENERAL COMMENTS	Background The authors give a good description of peer assessment and its strengths and barriers. What is the difference between peer assessment and peer learning? Should probably be highlighted in the background. Aim Four relevant questions are asked and answered. Method A strength in the method is doing the optional stage Consultation. However, the method chapter have several limitations; Study selection: Two researchers have not independently participated in all levels of a study selection. Only one author have screened titles and abstracts and read full-text for inclusion and exclusion. The text says “full-text articles were discussed with the other authors by the first author”. Critical appraisal: A scoping study seeks to present an overview of all material reviewed. Thus, doing critical appraisal is not recommended in a scoping review according to Arksey and O’Malley. Critical appraisal is the domain of a systematic review. Data extraction: Two authors should independently extract data from the first five to ten included studies using the data-charting form and meet to determine whether their approach to data extraction is consistent with the research question and purpose, according to Levac. This number is not accomplished in this scoping review. Furthermore, only 15 of 37 articles were critical appraised by two authors. Study limitations The rationale for choosing four databases is not discussed in study limitations. Could other databases broadened the field? This section is also missing a reflection of why doing critical appraisal in a scoping review. Some typo’s, i.e line 46, 289. Reference 12 and 19 are the same.
--

	Overall impression: Well-written text, informative tables, but limitations in the methodology for performing scoping reviews.
--	--

REVIEWER	Simon Schwill University Hospital Heidelberg Department of General Practice and Health Services Research Germany
REVIEW RETURNED	30-Oct-2020

GENERAL COMMENTS	Dear Dr. Sucksmith, dear Colleagues, Thank you very much for the opportunity to review this work. The manuscript summarizes the literature on formative peer assessment in higher education in healthcare. Design of the study is provided and both, results and methods are given. In summary, this manuscript is a well written para-systematic review which has required a lot of time to perform. I think it merits publication, however, some modifications are needed, especially in the discussion, limitations and outlook. Title - Fine. Abstract - Disc: "For formative peer assessment to be effective it needs to be implemented in a collaborative learning environment." Not covered by the data in the abstract nor the body, should be removed. - Disc: "Since peer collaboration seems to affect students' and teachers' experiences of formative peer assessment empirical investigations exploring collaboration between students is of utmost importance." Fine conclusion and rating of the authors. Methods - P 5, l 141/142 "(b)..." Missunderstanding to me: medicine not mentioned? Please clarify - Compare with Fig 1: exclusion: no undergrad?! Please clarify in the methods - Date of mesh term: May 29 th 2019: what was the period of time? 1900 – May 29th 2019? Please clarify Results - Fine to me. What I miss is the organizational costs and benefits of peer-assessment (see disc) Discussion - I like your presentation with the 2 steps and perfectly agree that aims of the peer-assessment should be transparent. What I do not agree with is that the purpose of peer-assessment should mono-rational and just for the good of it (to improve the collaborative learning). Please discuss: - (1) Peer-assessment helps to improve the educational skills for the peer-assessors themselves. By changing the point of view =slip into the assessors position they widen their understanding for the process of learning and assessment. (compare Burgess A, Clark T, Chapman R, Mellis C. Senior medical students as peer examiners in an OSCE. Med Teacher. 2013;35(1):58–62. AND Homberg A, Hundertmark J, Krause J, Brunnée M, Neumann B, Loukanova S. Promoting medical competencies through a didactic tutor qualification programme - a qualitative study based on the CanMEDS Physician Competency Framework. BMC Med Educ 2019;19:187. doi: 10.1186/s12909-019-1636-5) - (2) In high volume programs with 400 or 500 (medical) students per
--

	year (!) peer tutoring is the basis of small group learning and peer assessment is the basis for costly assessment such as the OSCE. This is the case for high volume programs (compare: Schwill et al. BMC Medical Education (2020) 20:17 https://doi.org/10.1186/s12909-019-1898-y) and even more for any higher education in healthcare out of our first world perspective!!! - Compare Page 10, L 306ff “Otherwise, there is a risk that students might perceive peer assessment as an activity meant to ease the burden on the teachers.” Please rephrase, to me it appears rather polemic. - Synthesis (comment): Only by being transparent on the educational and financial resources of the faculty as well as on the personal benefits for peers as assessments we treat our students the way to be treated: as individuals who are able to judge and decide for themselves! Limitations! -Please add that there was an end-date of the research period, and no studies after “x.x.2019” could be included Fig1 - Well presented. Appendix 1-2 - Well presented. Appendix 3 - Happy to see a PRSIMA-ScR checklist. I would love to see a revised version of the manuscript to be published in BMJ open. Best regards.
--	--

VERSION 1 – AUTHOR RESPONSE

reviewer # 1	Author Response	Changes made to article
The authors give a good description of peer assessment and its strengths and barriers. What is the difference between peer assessment and peer learning? Should probably be highlighted in the background.	Thank you for your thoughtful and insightful comments and suggestions. We have tried to respond to your comments. It is a wise question. However, we have chosen to focus peer assessment as an educational approach where feedback, communication, reflection, and collaboration between peers are key characteristics and as a learning activity in itself, regardless of educational model. We think that if we describe peer learning we also have to describe other peer collaborative models as for example, peer assisted learning, problem based learning, peer tutoring etc.	No changes made
Four relevant questions are asked and answered.	Thank you very much.	
Two researchers have not independently participated in all levels of a study selection. Only one author have screened titles and abstracts and read full-text for	We thank you for acknowledging this and agree that we have not expressed this clearly enough. We have tried to rephrase and hope that this may better describe the team contribution in the screening process (p. 6)	

inclusion and exclusion. The text says “full-text articles were discussed with the other authors by the first author”.		
A scoping study seeks to present an overview of all material reviewed. Thus, doing critical appraisal is not recommended in a scoping review according to Arksey and O’Malley. Critical appraisal is the domain of a systematic review.	Thank you for your comment on the critical appraisal. We do agree, Arksey and O’Malley do not recommend critical appraisal in a scoping review. However, in the recent debate concerning critical appraisal in scoping reviews we refer to Dautb (2013) saying; “Some authors have expressed their concerns about Arksey and O’Malley’s framework’s inability to provide for an assessment of the quality of the literature [1,2,3,4,]. We believe assessing for quality is a necessary component of scoping studies. The assessment itself is a significant task and should be performed using validated instruments.” (Daudt et al. BMC Medical Research Methodology 2013, 13:48) References in Daudt 2013  1. Anderson S, Allen P, Peckham S, Goodwin N: Asking the right questions: scoping studies in the commissioning of research on the organization and delivery of health services. Health Research Policy and Systems 2008, 6:1–12. 2. Davis K, Drey N, Gould D: What are scoping studies? A review of the nursing literature. Int J Nurs Stud 2009, 46:1386–1400. 3. Brien SE, Lorenzetti DL, Lewis S, Kennedy J, Ghali WA: Overview of a formal scoping review on health system report cards. Implementation Science 2010, 5:2. 4. Levac D, Colquhoun H, O’Brien KK: Scoping studies: advancing the methodology. Implementation Science 2010, 5:69. We have added an argument for justifying the critical appraisal in a scoping review. (p. 6)	p. 6
Two authors should independently extract data from the first five to ten included studies using the data-charting form and meet to determine whether their approach to data extraction is consistent with the research question and purpose, according to Levac. This number is not accomplished in this scoping review. Furthermore, only 15 of 37 articles were critical appraised by two authors.	Thank you and we agree that this may be seen as a limitation in a scoping review. We have tried to clarify how data was extracted by using the developed charting form (appendix 2) as a basis in our critical discussion among the team members (p 6, 7).	p.6, 7
The rationale for choosing four databases is not discussed in study limitations. Could other databases broadened the field?	Thank you, we have tried to be more precise in the method section how the databases were chosen and added the consultation with a librarian (p 5) We have also addressed this as a limitation of the study (p.12).	p. 5, 12

This section is also missing a reflection of why doing critical appraisal in a scoping review.	We have made some clarification concerning our reason for conducting a critical appraisal. (p.6)	p.6
Some typo's, i.e line 46, 289. Reference 12 and 19 are the same.	Thank you for your attention. Space between words and references has been corrected.	p. 2 and 10 p.4
Well-written text, informative tables, but limitations in the methodology for performing scoping reviews.	Thank you. We respectfully acknowledge your valuable comments and hope that you will find our amendments to the methodology section satisfying.	
Reviewer #2	Author Respond	Changes made to article
Title -Fine.	Thank you for your thoughtful and insightful comments and suggestions. We have tried to respond to your comments.	
Abstract -Disc:" For formative peer assessment to be effective it needs to be implemented in a collaborative learning environment." Not covered by the data in the abstract nor the body, should be removed.	Thank you. We do agree and has removed the statement from the discussion in the abstract. (p 2)	p.2
Disc: "Since peer collaboration seems to affect students' and teachers' experiences of formative peer assessment empirical investigations exploring collaboration between students is of utmost importance." Fine conclusion and rating of the authors.	Thank you for the comment.	
Methods -P 5, l 141/142 "(b)..." Missunderstanding to me: medicine not mentioned? Please clarify	We are very grateful for your attention and has revised the phrasing of medicine correctly. (p.5)	p.5
Compare with Fig 1: exclusion: no undergrad?! Please clarify in the methods	We have revised the phrasing of exclusion in figure 1 for clarification.	Figure 1
Date of mesh term: May 29 th 2019: what was the period of time? 1900 – May 29th 2019? Please clarify	We did not have any limitations of the period; all indexed studies in each database were included until the last search in May 28 th 2019. We have clarified this in the method section. (p. 6)	6
Results -Fine to me. What I miss is the organizational costs and benefits of peer-assessment (see disc)	Thank you. For the legibility, and as presented at page 8, line 228, we chose to present the result of the context, rational and interventions for formative assessment in a table (3). However, we do agree with your comment and have tried to highlight the cost benefits in the discussion (p. 10 and 11). However, cost benefits are not thoroughly discussed in the included studies and we have only chosen to add cost benefits in our discussion.	10 and 11
Discussion	Thank you for your comments and suggestions	10

I like your presentation with the 2 steps and perfectly agree that aims of the peer-assessment should be transparent. What I do not agree with is that the purpose of peer-assessment should mono-rational and just for the good of it (to improve the collaborative learning). Please discuss: -(1) Peer-assessment helps to improve the educational skills for the peer-assessors themselves. By changing the point of view =slip into the assessors position they widen their understanding for the process of learning and assessment. (compare Burgess A, Clark T, Chapman R, Mellis C. Senior medical students as peer examiners in an OSCE. Med Teacher. 2013;35(1):58–62. AND Homberg A, Hundertmark J, Krause J, Brunnée M, Neumann B, Loukanova S. Promoting medical competencies through a didactic tutor qualification programme - a qualitative study based on the CanMEDS Physician Competency Framework. BMC Med Educ 2019;19:187. doi: 10.1186/s12909-019-1636-5)	from recent studies. We have tried to emphasize the importance of more than the collaborative part in our discussion by highlighting self-reflection, communication and knowledge development important in healthcare professions. We have also used your suggestions for reference from Homberg et.al 2019. (p 10).	
2) In high volume programs with 400 or 500 (medical) students per year (!) peer tutoring is the basis of small group learning and peer assessment is the basis for costly assessment such as the OSCE. This is the case for high volume programs (compare: Schwill et al. BMC Medical Education (2020) 20:17 https://doi.org/10.1186/s12909-019-1898-y) and even more for any higher education in healthcare out of our first world perspective!!!	We appreciate your recommendation and we have highlighted the cost benefits in our manuscript and refers to Schwill et al 2020 (p. 11).	p. 11
Compare Page 10, L 306ff “Otherwise, there is a risk that students might perceive peer assessment as an activity meant to ease the burden on the teachers.” Please rephrase, to me it appears rather polemic.	We have removed the comment and reference (p.10)	10
Synthesis (comment): Only by being transparent on the educational and financial resources of the faculty as well as on the personal benefits for peers as assessments we treat our students the way to be treated: as individuals who are able to judge and decide for themselves! Limitations!	Thank you, we have included this as a limitation (p.12).	p.12

-Please add that there was an end-date of the research period, and no studies after "x.x.2019" could be included		
Fig1 -Well presented. Appendix 1-2 -Well presented. Appendix 3 -Happy to see a PRSIMA-ScR checklist.	Thank you.	

VERSION 2 – REVIEW

REVIEWER	Marit Hegg Reime Western Norway University of Applied Science, Norway.
REVIEW RETURNED	30-Nov-2020

GENERAL COMMENTS	I think you have done a good job with the revision of the paper and have answered out the earlier review comments.
--

REVIEWER	Simon Schwill University Hospital Heidelberg, Germany
REVIEW RETURNED	13-Dec-2020

GENERAL COMMENTS	Thank you very much for the opportunity to review the revised version of the manuscript. You have extensively worked on the manuscript and integrated the reviewer's comments. I think the work merits publication and is a good overview about the current situation.
---